# Study of the Corrosion Mechanism of Iron-Based Amorphous Composite Coating with Alumina in Sulfate-Reducing Bacteria Solution

Zhenhua Chu [1,*], Haonan Shi [1], Fa Xu [1], Jingxiang Xu [1], Xingwei Zheng [2], Fang Wang [1], Zheng Zhang [1] and Qingsong Hu [1]

1 Department of Mechanical Engineering, College of Engineering, Shanghai Ocean University, Shanghai 201306, China

2 College of Science, Donghua University, Shanghai 201620, China

* Correspondence: zhchu@shou.edu.cn

**Abstract:** In this work, a composite coating composed of iron-based amorphous material and alumina mixed with 13 wt.% titanium oxide (AT13) ceramic was successfully fabricated by High Velocity Air-fuel Flame Spray (HVAF). The corrosion process of the composite coating in Sulfate-Reducing Bacteria (SRB) solution for 31 d was investigated by Electrochemical Impedance Spectroscopy (EIS). The corrosion morphologies and corrosion products were tested by X-ray photoelectron spectroscopy. The corrosion mechanism can be divided into two stages: microbial adhesion and biofilm failure. The microbial adhesion on the surface of the composite coating improved the formation of biofilm, which improved the corrosion resistance. On the other hand, the SRB metabolic process in the biofilm accelerated the formation of corrosion products, which resulted in the failure of the biofilm and thus the composite coating was re-exposed in the corrosion solution.

**Keywords:** iron-based amorphous coating; ceramic; corrosion; SRB





## 1. Introduction

The marine environment is a complex corrosion environment. In addition to the corrosion damage by seawater medium, the widespread microorganisms in the marine environment can also affect the corrosion behavior of alloys [1,2]. Generally, the service failure of metal materials can be caused by microbiologically influenced corrosion (MIC), due to microorganisms and their metabolic activities [3–6]. This is an important type of corrosion, causing the failure of marine engineering materials. The SRB is one kind of well-known anaerobic bacterium existing widely in the deep sea and causing MIC. Researchers investigated the polarization resistance of stainless steel in a sterile solution and SRB solution, and the results indicated that the corrosion rate of the coupons in SRB solution was 10 times higher than that in sterile medium [7,8]. This is caused by the uneven adsorption of SRB biofilm on the surface of metal materials [9]. Therefore, the corrosion mechanism caused by SRB and protection technologies for severe corrosion environments require further study.

Coating technology is regarded as one of the effective approaches for the protection of steels from corrosion [10–12]. Water-borne coatings have been used widely in the last decade due to the provisions of low volatile organic compound emissions [13,14]. However, some problems also need to be noted, such as toxicity, carcinogenicity and the pollution of biocides, accompanying the release of antibacterial agents to the environment [15,16]. On the other hand, the interaction between the abrasion caused by sea mud and corrosion will accelerate the failure of materials. Thus, it is promising to fabricate a novel coating with high abrasion and corrosion resistance.

The high strength, high hardness and superior corrosion resistance of iron-based amorphous alloys have attracted the attention of researchers [17–20]. However, the poor

glassy formation ability of amorphous alloy restricts its applications. In order to avoid the disadvantages of iron-based amorphous alloys, we can fabricate amorphous coatings using thermal spraying technology [21]. Zhou et al. [22] prepared a dense iron-based amorphous coating by high-speed oxygen fuel spraying. The electrochemical test results showed that the coating exhibited excellent corrosion resistance in 3.5% sodium chloride, 1 N hydrogen chloride and 1 N sulfuric acid solution. Chu [23,24] fabricated iron-based amorphous composite coatings with AT13 and titanium nitride with high corrosion resistance and wear resistance. However, the corrosion resistance of amorphous composite coating in SRB solution needs to be further studied.

In the present work, the corrosion resistance of iron-based amorphous coatings with AT13 in SRB solution was studied systematically. The electrochemical characteristics of iron-based amorphous composite coating and EIS were investigated. The corrosion mechanism caused by SRB was proposed.

## 2. Materials and Experimental Process

Fe$_{54}$Cr$_{25}$Mo$_{17}$C$_2$B$_2$ alloy amorphous powders were produced by high pressure Ar gas atomization. Then, sprayed AT13 ceramic powders (20–40 μm) and amorphous powders were mixed by a mechanical mixing machine for 4 h. The sprayed powders are shown in Figure 1. Mild steel (0.45 wt.% C) was selected as the substrate with a size of 10 mm × 10 mm × 12 mm. HVAF was adopted to fabricate iron-based amorphous coatings and composite coatings. The parameters of the spraying process by HVAF are summarized as shown in Table 1.

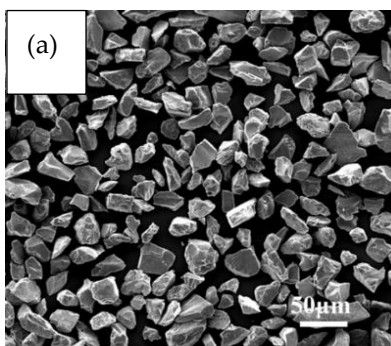 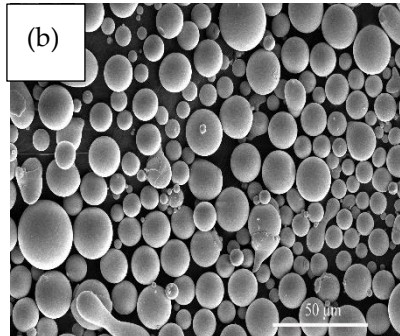

**Figure 1.** Sprayed powder morphologies: (**a**) AT13; (**b**) Fe-based amorphous powders.

**Table 1.** Spraying parameters of the HVAF process.

| Coating by HVAF | Parameter |
|---|---|
| Spray distance (mm) | 180 |
| Air pressure (MPa) | 0.54 |
| Fuel 1 press (MPa) | 0.48 |
| Fuel 2 press (MPa) | 0.26 |
| Powder delivery rate (rpm) | 3 |

The morphologies of coatings were observed by scanning electron microscopy (SEM, S4800, Hitachi, Tokyo, Japan). X-ray diffraction (XRD, Bruker D8 Focus, Billerica, MA, USA) was adopted to analyze the microstructures of coatings. A fluorescence microscope was used to observe the adhesion of microorganisms on the coating surface. Porosity data were determined according to the results of the SEM photos and Image-Plus software. More than 6 images were chosen.

Electrochemical tests were performed by a three-electrode cell, including a saturated calomel electrode (SCE), a graphite electrode as the reference and an auxiliary electrode. Specimens for the corrosion test were closely sealed with epoxy resin, leaving only an end-surface with a surface area of 1 × 1 cm$^2$ exposed for testing. A Tafel plot was created at a potential sweep rate of 0.5 mVs$^{-1}$ from −100 mV to 1500 mV in SRB solution, which was

open to air after immersing the specimens for an hour. In addition, EIS was examined in SRB solution. The impedance plots were interpreted on the basic of the equivalent circuit using a suitable fitting procedure by Echem Analyst. After EIS measurement, the corroded surface was examined by SEM. The corrosion products also were investigated by X-ray photoelectron spectroscopy.

In this study, the SRB were purchased from Beina Chuanglian Biotechnology Co., Ltd., Beijing, China. SRB medium included 0.98 g $MgSO_4$, 1.0 g yeast extract, 1.0 g $NH_4Cl$, 1.0 g $Na_2SO_4$, 0.5 g $K_2HPO_4$, 0.1 g $CaCl_2 \cdot 2H_2O$, 1.0 mg Resazurin, 0.5 g $FeSO_4 \cdot 7H_2O$, 0.1 g Na-thioglycolate, and 1 L deionized water. The PH of the medium was adjusted to $7.8 \pm 0.2$ by NaOH. The growth curve of SRB was measured by an ultraviolet spectrophotometer; see Figure 2. It can be seen from the figure that 0~5 h was the slow period, and 6~16h was the logarithmic growth period. The stable period occurred after 16 h. In this experiment, all samples were soaked in SRB solution when in the stable growth period. Meanwhile, fresh bacterial liquid was replaced every three days to ensure the activity of bacteria.

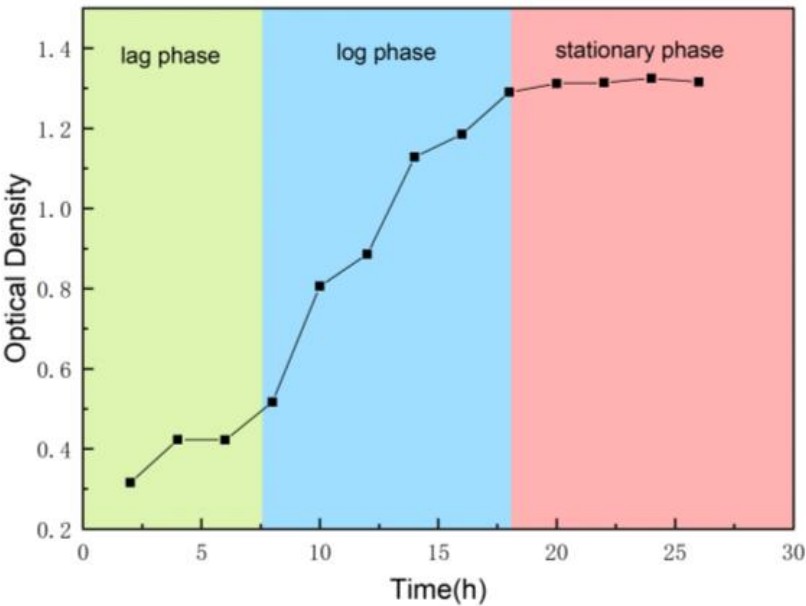

**Figure 2.** The growth curve of SRB.

## 3. Results

The as-sprayed iron-based amorphous coating and composite coatings with various contents of AT13 were tested by X-ray diffraction, and the XRD patterns are shown in Figure 3a. For the iron-based amorphous coating, there was a broad diffraction bump at the angle of $2\theta = 30–43°$. This represented the amorphous phase. For the composite coatings, the Bragg peaks were associated with AT13 crystalline phases, which indicates that the composite coating was composed with amorphous and AT13. The iron-based amorphous composite coating could be successfully prepared by HVAF technology.

The cross-section of the composite coating with iron-based amorphous and 15 wt.% AT13 is shown in Figure 3b. It is found that the as-sprayed coating is closely bonded to the substrate and the AT13 particles are distributed homogeneously; the composite coatings are dense structures. All of coatings had a thickness of about 300 μm.

The porosities of coatings were calculated based on the SEM photos. The statistical results are shown in Figure 4b. With the addition of AT13 into iron-based amorphous coating, the porosity is reduced. Meanwhile, the smallest porosity is obtained for the composite coating with 15% AT13.

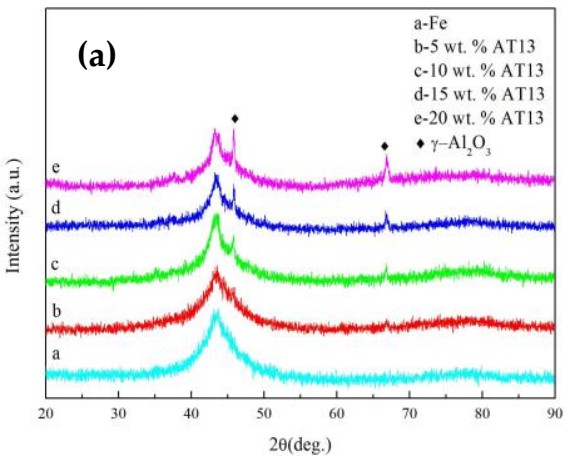
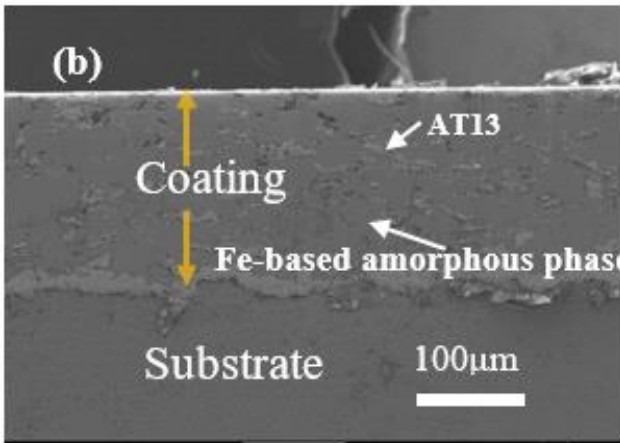

**Figure 3.** (**a**) XRD patterns of composite coatings with various content of AT13; (**b**) surface morphology of iron-based amorphous coating with 15 wt.% AT13.

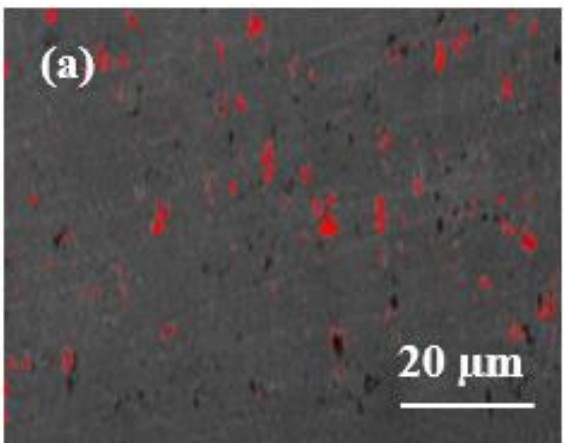
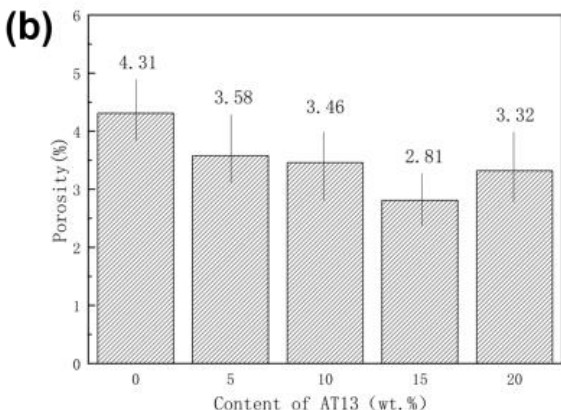

**Figure 4.** (**a**) SEM photo of coatings for calculation of porosity and (**b**) statistical porosities of coatings.

Firstly, the potendiodynamic polarization curves of different samples with various content AT13 composite coatings were investigated as shown in Figure 5. Some electrochemical parameters, such as corrosion potential ($E_{corr}$), corrosion current density ($i_{corr}$), transpassive current density ($i_{pass}$), transpassive potential ($E_{tr}$) and corrosion rate, are listed in Table 2. It clearly shows that the corrosion resistance of composite coatings is improved with the introduction of AT13. The $E_{corr}$ of the composite coating is larger than that of iron-based amorphous coating, and the $i_{corr}$ of the composite coating is lower than that of iron-based amorphous coating. Meanwhile, it is worthy to note that the composite coating with 15 wt.% AT13 has the largest corrosion potential and transpassive potential. This indicates that the best corrosion resistance was obtained for the composite coating with 15 wt.% AT13.

Furthermore, EIS measurement was adopted to evaluate the corrosion failure process, and the immersion experiments in SRB solution combined with EIS measurements were carried out to analyze the corrosion behavior of the coating. Figure 6 illustrates the EIS plots of the coatings with 15 wt.% AT13 compared with iron-based amorphous coatings immersed in SRB solution for 31 d. The Nyquist plots are shown in Figure 6a,c. It is shown that the capacitive arc decreases firstly in a small range for 1d. It is considered that the corrosive solution gradually penetrated into the coating from the pores of the surface. This indicates the corrosion resistance is reduced at the beginning.

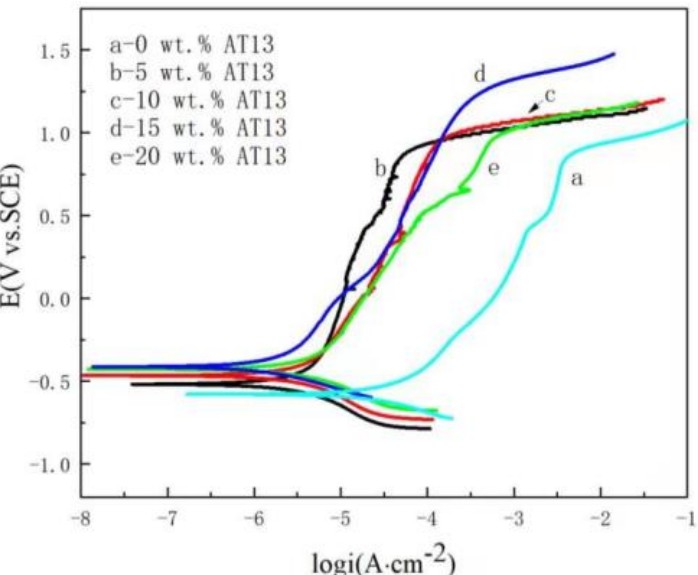

**Figure 5.** Potentiodynamic polarization curves of different samples with various contents of AT13.

**Table 2.** Electrochemical parameters obtained from potentiodynamic polarization curves of coatings.

| Content of AT13/wt.% | $E_{corr}$/mV | $i_{corr}$/A cm$^{-2}$ | $i_{pass}$/A cm$^{-2}$ | $E_{tr}$/mV | Corrosion Rate/mpy |
|---|---|---|---|---|---|
| 0 | −580 | $5.14 \times 10^{-5}$ | $5.42 \times 10^{-3}$ | 867 | 25.06 |
| 5 | −519 | $7.50 \times 10^{-6}$ | $6.23 \times 10^{-5}$ | 905 | 6.47 |
| 10 | −467 | $5.01 \times 10^{-6}$ | $1.12 \times 10^{-4}$ | 993 | 5.49 |
| 15 | −411 | $1.75 \times 10^{-6}$ | $5.37 \times 10^{-4}$ | 1236 | 2.07 |
| 20 | −430 | $4.06 \times 10^{-6}$ | $6.94 \times 10^{-4}$ | 1012 | 4.60 |

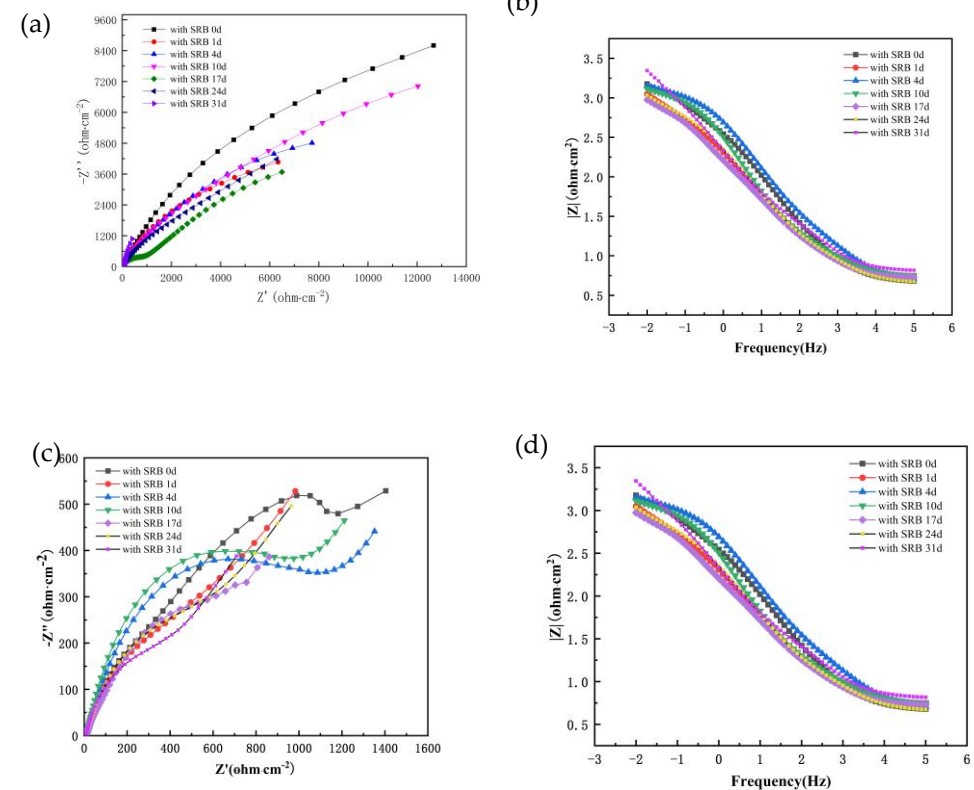

**Figure 6.** Electrochemical impedance spectrogram of Fe-based amorphous coating soaked in SRB solution: (**a**) Nyquist plot; (**b**) Bode plot; and composite coating with 15 wt.% AT 13 soaked in SRB solution: (**c**) Nyquist plot; (**d**) Bode plot.

However, the radius of the capacitive reactance arc for both the iron-based amorphous coating and the composite coating increased after being immersed for 4 d and 10 d. Meanwhile, for the composite coating with AT13, the radius of the capacitive reactance arc was larger than that of the initial stage. The largest radius was obtained when the soak time was 10 d. Generally, the large radius of the capacitive reactance arc means high corrosion resistance. Thus, it is suggested that the corrosion resistance is improved after immersion for 10 d. The arc radius of high-frequency capacitive reactance is related to the charge transfer resistance and corrosion product film. The increase of capacitive reactance arc radius indicates that the charge transfer resistance is increased, which means the protective ability of the corrosion product film is enhanced. On the other hand, it is worthy to note that Warburg impedance can be observed in the composite coating. this is caused by the diffusion of corrosive ions in solution. Meanwhile, the Bode impedance plots of the composite coating show two continuous semi-circle shapes for 4 d and 10 d (as shown in Figure 6d). This means that there are two time constants. One CPE time constant is in the high frequency range related to the capacitance impedance of the coating. The other constant is the Warburg impedance caused by the reaction metabolic process of SRB.

As immersion time increased, the radius of capacitance arc was reduced gradually after 17 d. The smallest radius of the capacitance arc was obtained for the iron-based amorphous coating. This indicates more defects are formed in the coating due to corrosion. The other time constant in the Bode plot in the low frequency range is related to charge transfer resistance and a double-layer capacitance between the solution/coating interface.

According to the Bode plot, the EIS curves are fitted by the proposed equivalent circuit. There are two stages for the corrosive process. For the initial stage, the equivalent circuit model (Figure 7a) is used, which is composed of the resistance of electrolyte solution ($R_s$), the CPE of coating ($Q_c$) in parallel with the resistance of coating ($R_{ct}$), and inductive reactance W [25–27]. Model B (Figure 7b) is applied to analyze the later corrosion stage of the composite coatings. For model B, CPE-cf represents the characteristics of the external film layer and film resistance ($R_f$). The EIS fitting parameters of various components are summarized in Table 3. The $R_{ct}$ is decreased firstly and then increased. Finally, it is reduced. This indicates the initial increase of corrosion resistance. The biofilm is broken, so the electrolyte solution penetrates into the coating from the pores.

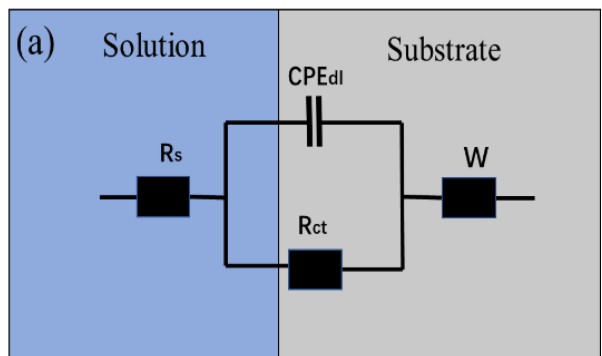
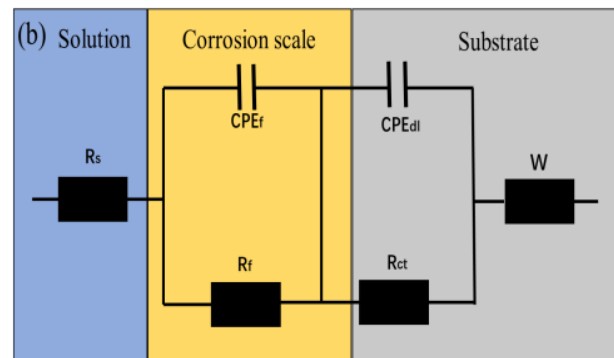

**Figure 7.** Schematic diagram of the equivalent circuit: (**a**) the pre-corrosion stage of the coating (0~10 d); (**b**) the corrosion stage of the coating (17~31 d).

**Table 3.** Summary of the EIS fitting parameters for various components.

| Time (day) | $R_s$ ($\Omega \cdot cm^{-2}$) | $R_{ct}$ ($\Omega \cdot cm^{-2}$) | $CPE_{dl}$ | $R_f$ ($\Omega \cdot cm^{-2}$) | $CPE_f$ | Goodness of Fit |
|---|---|---|---|---|---|---|
| 0 | 4.56 | 950.1 | $8.58 \times 10^{-4}$ | / | / | $2.20 \times 10^{-3}$ |
| 1 | 5.036 | 402.1 | $2.07 \times 10^{-3}$ | / | / | $1.13 \times 10^{-3}$ |
| 4 | 3.845 | 838.3 | $5.05 \times 10^{-4}$ | / | / | $1.08 \times 10^{-3}$ |
| 10 | 5.026 | 680.4 | $7.94 \times 10^{-4}$ | / | / | $1.09 \times 10^{-3}$ |
| 17 | 4.992 | 177.1 | $8.80 \times 10^{-3}$ | 588.7 | $3.14 \times 10^{-3}$ | $3.19 \times 10^{-4}$ |
| 24 | 4.492 | 133.4 | $4.23 \times 10^{-4}$ | 489.8 | $1.46 \times 10^{-3}$ | $4.04 \times 10^{-5}$ |
| 31 | 2.781 | 101.6 | $1.2 \times 10^{-1}$ | 274.8 | $2.04 \times 10^{-3}$ | $6.47 \times 10^{-4}$ |

According to the results of the EIS, the corrosion process can be divided into two stages: the pre-corrosion stage and the corrosion stage. When the sample is immersed in the SRB solution for 1d, the adhesion of some microorganisms on the surface can be observed, as marked in Figure 8a. With the increased immersion time, the microbial film is formed with the metabolic process. So, the Warburg impedance appears in the EIS results due to the transfer of charge. On the other hand, the microbial film protects the substrate from corrosion. Therefore, the capacitive reactance arc radius is increased from 1 d to 10 d.

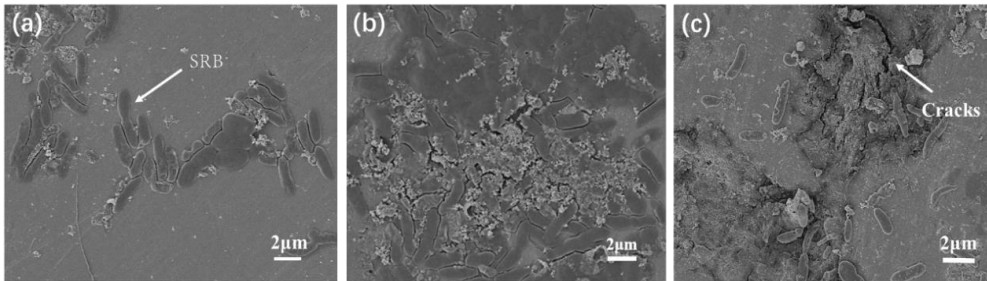

**Figure 8.** Corrosion morphology of AT13 composite coating: (**a**) soak for 1d; (**b**) soak for 17 days; (**c**) soak for 31 days.

However, when the immersion time exceeds 17 d, the broken microbial film can be observed as shown in Figure 8b. The corrosion resistance is reduced. Meanwhile, the capacitance is formed between the microbial film and the composite coating. The corrosive solution penetrates into the composite coating. When the immersion time is 31 d, the microbial film is peeled, and the crack on the composite coating is observed, as shown in Figure 8c.

Meanwhile, the corrosive surface of the composite coating was also observed by fluorescence microscope after 1 d, 17 d and 31 d, as shown in Figure 9. The green fluorescence represents the adhesion of SRB on the surface. It can be found that there are some SRBs on the coating surface when it is immersed for one day. With the increase of immersion time, more and more SRBs appear on the surface of the coating. When the sample is immersed for 31 d, some colonies are formed on the surface of the coating.

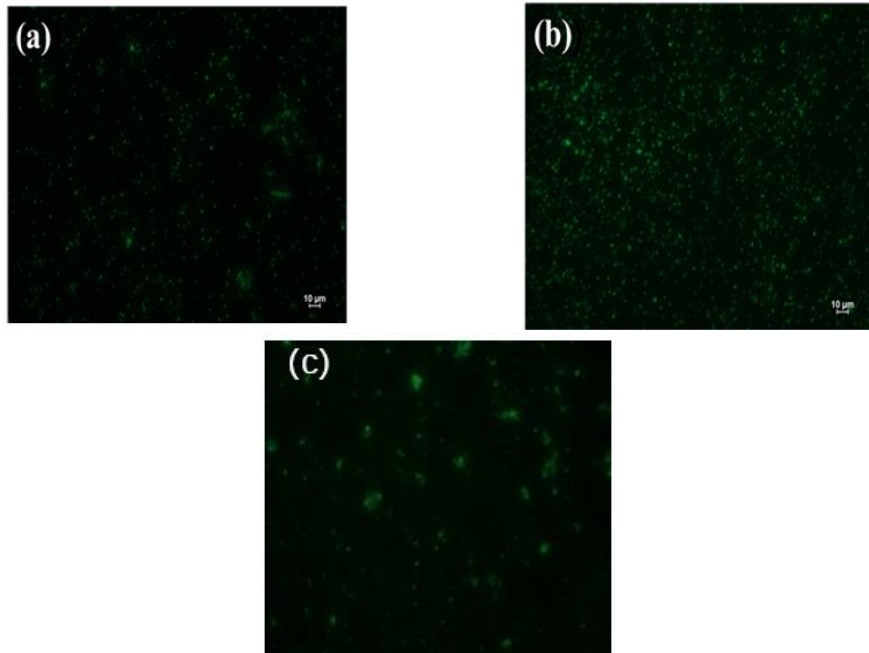

**Figure 9.** Adhesion of SRB on the surface of the composite coating after immersion for (**a**) 1 d, (**b**) 17 d, (**c**) 31 d.

The corrosion products were analyzed by X-ray photoelectron spectroscopy (XPS). Figure 10a shows the main distribution of sulfur ions in the corrosion products. S comes from the metabolite of SRB bacteria. The sulfur element is composed of four peaks, mainly $SO_4^{2-}$ and $S^{2-}$. The peak area of $S^{2-}$ is larger than that of $SO_4^{2-}$. According to the BCSR theory, SRB will consume $SO_4^{2-}$, and the final product is $HS^-$; the reactions are as follows [25,26]:

$$H_2O \rightarrow 2H^+ + OH^- \tag{1}$$

$$H^+ \xrightarrow{Hydrogenase} [H] \tag{2}$$

$$SO_4^{2-} + 8[H] \xrightarrow{SRB} 4H_2O + S^{2-} \tag{3}$$

where [H] acts as an electron transfer medium, which is generated by the cathode reaction. During the metabolic process of the SRB, the film is formed on the surface of the composite coating.

However, the metabolic process enhances the acidity covering the coating at the same time. This results in the corrosion of the coatings. The Fe element is transferred to $Fe^{2+}$ and $Fe^{3+}$ as follows:

$$Fe \rightarrow Fe^{2+} + 2e^- \tag{4}$$

$$Fe \rightarrow Fe^{3+} + 3e^- \tag{5}$$

$$Fe^{3+} + 3OH^+ \rightleftharpoons Fe(OH)_3 \tag{6}$$

$$Fe^{2+} + 2OH^+ \rightleftharpoons Fe(OH)_2 \tag{7}$$

$$Fe^{2+} + S^{2-} \rightarrow 2FeS \tag{8}$$

FeS is formed with the metabolic product of the SRB. The XPS spectrum of Fe, $Fe(OH)_3$ and FeS are mainly formed as shown in Figure 10b.

Meanwhile, there are three peaks in the XPS spectrum of the Al element, as shown in Figure 9c. They are $Al_2O_3$, $Al_2S_3$ and Al $(OH)_3$. The reactions are as follows:

$$Al^{3+} + S^{2-} \rightarrow Al_2S_3 \tag{9}$$

$$Al^{3+} + 3OH^+ \rightarrow Al(OH)_3 \tag{10}$$

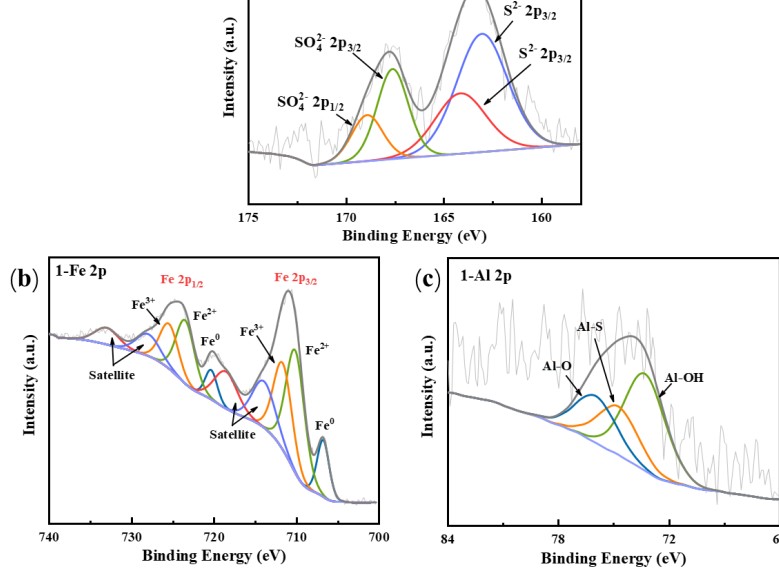

**Figure 10.** XPS atlas of corrosion product elements: (**a**) S; (**b**) Fe; (**c**) Al.

## 4. Discussion

At present, researchers have proposed various theories to explain the corrosion mechanism of SRB [27–29]. One of them is the cathode depolarization theory (CDT), which reported that the corrosion is caused by utilizing a hydrogenase to consume the hydrogen produced by cathodic reactions [30]. Another one is the mechanism of sulfides produced by the metabolism of SRB [31]. The biocatalytic cathodic sulfate reduction (BCSR) theory was proposed and developed by Gu [32] and Xu et al. [33]. The studies showed that SRB can induce MIC by acting as the acceptor of the electrons produced by anodic reactions (metal dissolution), which generally prevails under the conditions lack of carbon source.

According to above results, the corrosion mechanism is proposed. The schematic of the corrosion mechanism is shown in Figure 11. In the SRB environment, [H] is as a medium for electron transfer, so the corrosion of SRB causes a large amount of H to leave the composite surface, accelerating the cathode reaction. At the same time, the anode reaction remains stable, resulting in a large number of electrons being consumed on the composite surface. This indicates that the reaction through which SRB consumes [H] further promotes the metabolic activity of SRB. Meanwhile, the biofilm is formed, which can effectively block the contact between corrosive factors in the solution. However, when the corrosion products in the film cannot be metabolized, this leads to the deterioration of the film environment. At the same time, the lack of nutrients in the film will also cause the bacteria in the biofilm to start to take electrons from the composite coating. Then, with the acidity increase, the SRB corrosion produces porous FeS, which accelerates the absorption of H and the cathodic depolarization process, accelerating the corrosion rate. Finally, the film cracks due to the change of the internal environment, re-exposing the composite coating in the solution.

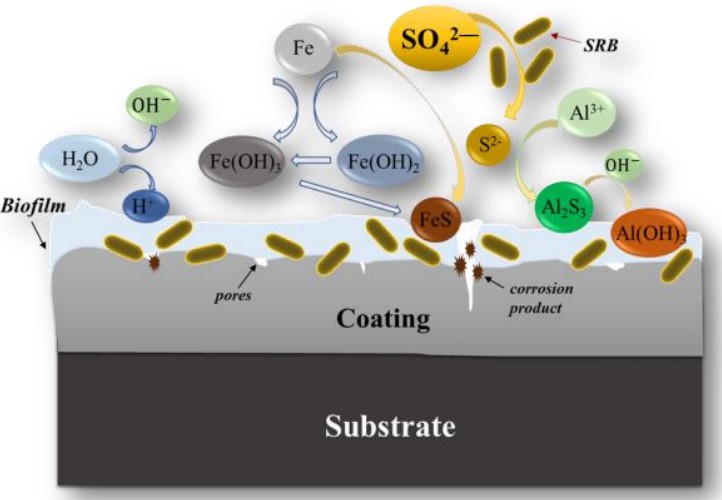

**Figure 11.** The schematic of the corrosion mechanism.

## 5. Conclusions

The composite coatings with iron-based amorphous and AT13 were prepared by HVAF. The corrosion process in SRB bacteria was studied by EIS. The corrosion products were analyzed by SEM and XPS. Based on the results, the corrosion mechanism was proposed. The SRB corrosion process can be divided into two stages: one is microbial adhesion, which is accompanied with SRB metabolic process and biofilm is formed. The other one is failure of the biofilm. With the acidity increase, the corrosion produces porous FeS and $Al(OH)_3$ increase, which accelerates the absorption of H and the cathodic depolarization process, accelerating the corrosion rate. Finally, the biofilm is destroyed and the composite coating is re-exposed in the solution.

**Author Contributions:** Formal analysis, J.X., X.Z. and Z.Z.; Investigation, Z.C. and Q.H.; Data curation, H.S., F.X. and F.W.; Writing—original draft preparation, Z.C. and F.X.; Writing—review and editing, Z.C. and H.S. All authors have read and agreed to the published version of the manuscript.

**Funding:** The present work was supported by the National Nature Science Foundation of China (Grant No. 51872072). The authors would like to express their gratitude for the support of the Fishery Engineering and Equipment Innovation Team of Shanghai High-level Local University, the State Key Laboratory for Mechanical Behavior of Materials, the program for Shanghai Collaborative In-novation Center for Cultivating Elite Breeds and Green-culture of Aquaculture animals (No. 2021-KJ-02-12) and the Natural Science Foundation of Shanghai (No. 20ZR1424000).

**Institutional Review Board Statement:** Not applicable.

**Informed Consent Statement:** Not applicable.

**Data Availability Statement:** Not applicable.

**Conflicts of Interest:** The authors declare no conflict of interest.

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
