# Peer review of "Study of the Corrosion Mechanism of Iron-Based Amorphous Composite Coating with Alumina in Sulfate-Reducing Bacteria Solution"

_coatings, doi:10.3390/coatings12111763_

Round 1

Reviewer 1 Report

In this manuscript, Chu et al. described the “Study of the corrosion mechanism of AT13/Fe-based amorphous composite coating in SRB solution” in detail. All the samples are properly characterized, and the morphology is excellent. However, still some problems exist which need to be addressed before the acceptance of this manuscript. I therefore advise minor review at this stage keeping in mind the following questions.

1) The English representation is very poor and must be thoroughly checked in the revised manuscript.

2) What is the stability of your samples as sulfur containing materials are usually unstable.

3) What is role of pH and how sulfur containing species maintain a buffer environment for bacterial inactivation.

4) Introduction is too short and needs proper attention with reasonable statements about the matter concerned.

5) both present and past tenses are used in the same paragraph. Usually, experimental work is described in past tense while explanation is usually carried out in present tense.

6) the following three papers needs proper citation in the XPS and EIS part of the manuscript.

i) N. Ali, A. Zada, M. Zahid, A. Ismail, M. Rafiq, A. Riaz, A. Khan, Enhanced photodegradation of methylene blue with alkaline and transition-metal ferrite nanophotocatalysts under direct sun light irradiation, J. Chin. Chem. Soc. 66 (2019) 402-408.

ii) H. Yasmeen, A. Zada, S. Liu, Surface plasmon resonance electron channeled through amorphous aluminum oxide bridged ZnO coupled g-C3N4 significantly promotes charge separation for pollutants degradation under visible light, J. Photochem. Photobiol. A: Chem. 400 (2020) 112681.

iii) A. Zada, Y. Qu, S. Ali, N. Sun, H. Lu, R. Yan, X. Zhang, L. Jing, Improved visible-light activities for degrading pollutants on TiO2/g-C3N4 nanocomposites by decorating SPR Au nanoparticles and 2,4-dichlorophenol decomposition path, J. Hazard. Mater. 342 (2018) 715-723.

Author Response

Responds to Reviewer 1

In this manuscript, Chu et al. described the “Study of the corrosion mechanism of AT13/Fe-based amorphous composite coating in SRB solution” in detail. All the samples are properly characterized, and the morphology is excellent. However, still some problems exist which need to be addressed before the acceptance of this manuscript. I therefore advise minor review at this stage keeping in mind the following questions.

1) The English representation is very poor and must be thoroughly checked in the revised manuscript.

● As suggested by the reviewer, it has been revised and carefully proof-read the manuscript to minimize spelling and grammatical errors.

2) What is the stability of your samples as sulfur containing materials are usually unstable.

● Specimens for the corrosion test were closely sealed with epoxy resin, leaving only an end-surface with a surface area of about 1 cm2 exposed for testing. And The sample is hung up and suspended in corrosive solution.

3) What is role of pH and how sulfur containing species maintain a buffer environment for bacterial inactivation.

● First of all, the PH of medium was adjusted to 7.8 by NaOH. Secondly, the samples were immersed in corrosion solution with 1L container for 3 samples. And the fresh bacterial liquid was replaced every three days to ensure the activity of bacteria. All of these is to maintain the PH is stable. In fact the PH in the real marine environment doesn’t change. Of course the PH in the local region maybe change due to the corrosion process. It should be studied deeply in our further study.As far as sulfur, it is metabolites products of Sulfate Reducing Bacteria. And it can combine with iron ions or aluminum ions to form corrosion products.

4) Introduction is too short and needs proper attention with reasonable statements about the matter concerned.

● As suggested by the reviewer, the Introduction has been modified in the revised manuscript.

5) both present and past tenses are used in the same paragraph. Usually, experimental work is described in past tense while explanation is usually carried out in present tense.

● Thanks for the suggestion and we have revised the errors carefully.

6) the following three papers needs proper citation in the XPS and EIS part of the manuscript.

i) N. Ali, A. Zada, M. Zahid, A. Ismail, M. Rafiq, A. Riaz, A. Khan, Enhanced photodegradation of methylene blue with alkaline and transition-metal ferrite nanophotocatalysts under direct sun light irradiation, J. Chin. Chem. Soc. 66 (2019) 402-408.

ii) H. Yasmeen, A. Zada, S. Liu, Surface plasmon resonance electron channeled through amorphous aluminum oxide bridged ZnO coupled g-C3N4 significantly promotes charge separation for pollutants degradation under visible light, J. Photochem. Photobiol. A: Chem. 400 (2020) 112681.

iii) A. Zada, Y. Qu, S. Ali, N. Sun, H. Lu, R. Yan, X. Zhang, L. Jing, Improved visible-light activities for degrading pollutants on TiO2/g-C3N4 nanocomposites by decorating SPR Au nanoparticles and 2,4-dichlorophenol decomposition path, J. Hazard. Mater. 342 (2018) 715-723

● These references have been added in the revised man

Reviewer 2 Report

This manuscript presents results of original experimental studies of corrosion in SRB (sulfate-reducing bacteria) solution of composite coating based on amorphous powder of Fe54Cr25Mo17C2B2 alloy and Al2O3-13%TiO2 (AT13) ceramics applied to mild steel samples. The results of studies of the course of corrosion of the coating by electrochemical impedance spectroscopy (EIS) showed that the process can be represented in the form of two subsequent stages: the stage of preliminary corrosion and the stage of corrosion. At the pre-corrosion stage, the formation of a microbial film on the surface of the coating as a result of the metabolic process of microorganisms was found to significantly reduce the corrosion rate. As the contact time of the SRB solution with the coating increases, the microbial film is destroyed and the corrosion rate increases. As a result, a cavity is formed between the microbial film and the composite coating, into which the corrosion solution penetrates.  The acidity of the solution increasing leads to the growth of porosity of the studied coating, that accelerates the rates of corrosion, rates of H uptake, and the catholic depolarization process.

The topic of the research is topical in scientific and applied respect.

The results presented in the manuscript are of interest to a wide range of specialists, graduate students and students developing biomedical equipment, marine equipment, including marine surface and underwater vessels, and equipment for port facilities, since microorganisms common in the marine environment, as shown by the research results, can affect the corrosion behavior of alloys.

The quality of the manuscript can be improved.

1) The manuscript should be supplemented with information on the parameters of the HVAF coating and its structure prior to corrosion testing. It is required to provide data not only on the average porosity, but also on the distribution of micro pore sizes in the coating, data on roughness parameters on the outer surface of the coating. These data are required to assess the possibility of changing the duration of the pre-corrosion stage. Increased roughness and the presence of open porosity contribute to the formation and preservation of the microbial film, especially in conditions of fluid movement.

2) The manuscript should be supplemented with information on the parameters of the coatings with different weight contents of AT13 for which tests were performed, presented in Figure 4. The effect of AT13 concentration in the coating on the pore distribution parameters in the coating, the roughness of the coating surface, and the thickness of the resulting coating should be discussed.

3) It is desirable to indicate a non-system designation of the duration of time in days («1d» Line 122 and further in the text). It is necessary to adhere to uniformity in the designation of time. Time designation is the "31 days" (Line 119), but the processing time indicates as "1d" (in line 122).

4) In the section 4 “Discussion” it is advisable to give a comparison of the results obtained with the results of similar studies of may be other types of coatings. This will emphasize the significance of the results obtained.

Author Response

The topic of the research is topical in scientific and applied respect.

The results presented in the manuscript are of interest to a wide range of specialists, graduate students and students developing biomedical equipment, marine equipment, including marine surface and underwater vessels, and equipment for port facilities, since microorganisms common in the marine environment, as shown by the research results, can affect the corrosion behavior of alloys.

The quality of the manuscript can be improved.

1) The manuscript should be supplemented with information on the parameters of the HVAF coating and its structure prior to corrosion testing. It is required to provide data not only on the average porosity, but also on the distribution of micro pore sizes in the coating, data on roughness parameters on the outer surface of the coating. These data are required to assess the possibility of changing the duration of the pre-corrosion stage. Increased roughness and the presence of open porosity contribute to the formation and preservation of the microbial film, especially in conditions of fluid movement.

● As suggested by the reviewer, the spraying parameters of the HVAF has been showed in the Table 1 in the revised manuscript.

   The porosity of the coating has been added as shown in Figure 4(b). The porosity was statistic value according to the results of the SEM photos and software of Image-Plus. And more than 6 images were chosen.

2) The manuscript should be supplemented with information on the parameters of the coatings with different weight contents of AT13 for which tests were performed, presented in Figure 4. The effect of AT13 concentration in the coating on the pore distribution parameters in the coating, the roughness of the coating surface, and the thickness of the resulting coating should be discussed.

● As pointed by the reviewer, the statistical results of the coatings porosity were shown in Figure 4 in the revised manuscript. Generally, the thickness of all kinds of coatings is the same, which is about 300 µm. The coatings before corrosion test were polished. So the roughness and the thickness are the same for all coatings.

3) It is desirable to indicate a non-system designation of the duration of time in days («1d» Line 122 and further in the text). It is necessary to adhere to uniformity in the designation of time. Time designation is the "31 days" (Line 119), but the processing time indicates as "1d" (in line 122).

● As pointed by the reviewer, the mistake has been corrected.

4) In the section 4 “Discussion” it is advisable to give a comparison of the results obtained with the results of similar studies of may be other types of coatings. This will emphasize the significance of the results obtained

● It is a good suggestion. In fact, we have tested aluminium alloys and magnesium alloy. The results are going to be published in our following paper. And as suggested by the reviewer, we tried to cited some other previous results to discuss.

Reviewer 3 Report

The manuscript "Coatings-2026768" by Chu et al. reported the Study of the corrosion mechanism of AT13/Fe-based amor-2 phous composite coating in SRB solution. After review, this study is interesting but the authors have to make major changes. The authors should refer to the following comments to improve their work:

General comments:

a. The language of the manuscript should be checked. There are many errors.

b. Correct reference format according to journal.

c. Please increase the number of references. The references seem to be purposefully following a series of articles with specific affiliations, please increase the scope of the references to avoid misunderstanding.

Specific comments:

Abstract:

a. Terms such as EIS, XPS, Al2O3, and TiO2 are abbreviated in the abstract without explanation, please revise and rewrite.

b. Please write the abbreviation after the explanation.

Introduction:

a. The first paragraph has no references, please revise and rewrite.

b. Please review the results of several other studies in the introduction and explain the difference between this study and the others.

c. Terms such as NaCl, HCl, H2SO4, and TiN are abbreviated in the introduction without explanation, please revise and rewrite.

d. Term SRB is explained in the abstract, please correct and rewrite.

Materials and methods:

a. The thickness of the layer (coating) is not specified. Please, give more information about the case.

b. Please add the amount used of each powder for the final product.

c. For corrosion test, the samples were sealed with epoxy resin, is epoxy suitable for this purpose? Can corrosive agents not penetrate the epoxy during the duration of this investigation (31 days)?

Results:

a. In the manuscript, it is said that Figure 3A is XRD patterns, but surface morphology is seen. Please revise and rewrite.

b. The XRD section is not acceptable at all. There is no scientific discussion, even the results are not reported correctly. This section needs a major revision. Please use the reference to explain the results.

c. If you want to show the sprayed coating (Figure 3), show the rest of the samples and compare.

d. The results obtained in polarization section are interesting, but unfortunately, there has been no discussion and it is only a data report. Prepare a table and report the polarization parameters such as the anodic Tafel slope, cathodic Tafel slope, the corrosion current density, corrosion rate, the percentage Inhibition efficiency, and polarization resistance for each sample. Please explain the results with scientific reasons.

e. The EIS section: Prepare a table and report the parameters such as RS (represents the solution resistance), RC (represents the coating resistance or polarization resistance), CPE (is the constant phase element which consists of two components i.e., QC (coating capacitance) and n (the coefficient related to the surface heterogeneity)), and log z.

f. In the discussion section, line 216: Reference number 37 is seen in a different format, while this manuscript has 14 references. Please, in case of using the results or discussion from any article, it should be cited in order to observe professional behavior.

g. In the conclusion section, line 223: "The long-term corrosion process..." Please remove "long-term". Investigating for 31 days cannot be considered long-term.

h. Please investigated the adhesion strength (pull-off adhesion tester) of the coatings.

Reviewer 4 Report

Report:

The article ''Study of the corrosion mechanism of AT13/Fe-based amor- 2 phous composite coating in SRB solution’’ is written well and I will recommend this manuscript after minor corrections/revision.

Points to be explained

11- Convert abbreviation AT13/Fe of the Title into full name.

22- There are few minor grammatical, spelling mistakes to be corrected.

33- Improve the Figures quality Fig-3b, Fig-4, Fig-5.

44- Compare results in the discussion part with the recent relevant literature.

Author Response

Responds to Reviewer

The article ''Study of the corrosion mechanism of AT13/Fe-based amorphous composite coating in SRB solution’’ is written well and I will recommend this manuscript after minor corrections/revision.

Points to be explained

11- Convert abbreviation AT13/Fe of the Title into full name.

  • It has been done.

22- There are few minor grammatical, spelling mistakes to be corrected.

  • As suggested by the reviewer, it has been revised and carefully proof-read the manuscript to minimize spelling and grammatical errors.

33- Improve the Figures quality Fig-3b, Fig-4, Fig-5.

  • It has been done.

44- Compare results in the discussion part with the recent relevant literature

  • As suggested by the reviewer, we tried to cited some other recent relevant results to discuss.

Round 2

Reviewer 3 Report

Thank you for the changes in the manuscript. Accept as it.